# Synaptic Pathology in Traumatic Brain Injury and Therapeutic Insights

**DOI:** 10.3390/ijms26199604

**Published:** 2025-10-01

**Authors:** Poojith Nuthalapati, Sophie E. Holmes, Hamada H. Altalib, Arman Fesharaki-Zadeh

**Affiliations:** 1Department of Neurology, Yale School of Medicine, New Haven, CT 06510, USA; poojith.nuthalpati@yale.edu (P.N.); sophie.holmes@yale.edu (S.E.H.); hamada.hamid@yale.edu (H.H.A.); 2Department of Neurology, West Haven Veterans Affairs Hospital, West Haven, CT 06516, USA; 3Department of Psychiatry, Yale School of Medicine, New Haven, CT 06510, USA

**Keywords:** traumatic brain injury, glutamate, synaptic plasticity, synaptic density, excitotoxicity, neuroinflammation

## Abstract

Traumatic brain injury (TBI) results in a cascade of neuropathological events, which can significantly disrupt synaptic integrity. This review explores the acute, subacute and chronic phases of synaptic dysfunction and loss in trauma which commence post-TBI, and their contribution to the subsequent neurological sequelae. Central to these disruptions is the loss of dendritic spines and impaired synaptic plasticity, which compromise neuronal connectivity and signal transmission. During the acute phase of TBI, mechanical injury triggers presynaptic glutamate secretion and Ca^2+^ ion-mediated excitotoxic injury, accompanied by cerebral edema, mitochondrial dysfunction and the loss of the mushroom-shaped architecture of the dendritic spines. The subacute phase is marked by continued glutamate excitotoxicity and GABAergic disruption, along with neuroinflammatory pathology and autophagy. In the chronic phase, long-term structural remodeling and reduced synaptic densities are evident. These chronic alterations underlie persistent cognitive and memory deficits, mood disturbances and the development of post-traumatic epilepsy. Understanding the phase-specific progression of TBI-related synaptic dysfunction is essential for targeted interventions. Novel therapeutic strategies primarily focus on how to effectively counter acute excitotoxicity and neuroinflammatory cascades. Future approaches may benefit from boosting synaptic repair and modulating neurotransmitter systems in a phase-specific manner, thereby mitigating the long-term impact of TBI on neuronal function.

## 1. Introduction

Traumatic Brain Injury (TBI), resulting from mechanic trauma of the brain parenchyma’s either penetrating or non-penetrating injury, is associated with a multitude of clinical manifestations such as neurocognitive sequelae being more prominent that encompass deficits in attention, memory and executive function [1,2], and these are frequently accompanied by psychiatric disturbances such as depression, anxiety and irritability [3]. Sensorimotor impairments may include weakness, ataxia, sensory loss and visual disturbances [4,5]. These symptoms arise from a primary mechanical injury (axonal shearing, contusions, hemorrhage) and a secondary cascade, leading to synaptic dysfunction and loss and persistent network disruption, underpinning both acute and chronic sequelae [6,7]. Hippocampal circuits and cortical networks show early synaptic degeneration after TBI [8]. Hippocampal circuits and cortical networks show early synaptic degeneration after TBI [9]. The Department of Veterans Affairs and Department of Defense guidelines highlight that both primary and secondary injury processes contribute to the clinical spectrum of TBI [10]. TBI precipitates a loss of synaptic integrity within the central nervous system (CNS) and initiates a cascade of pathophysiological changes triggered as a direct result of the primary biomechanical injury. The mechanical insult is followed by secondary processes involving neuroinflammation and oxidative injury, excitotoxicity and progressive synaptic degradation [11]. As a result, TBI survivors are predisposed to a significantly higher risk of developing neurological disorders, including Alzheimer’s disease, Chronic Traumatic Encephalopathy (CTE), epilepsy and movement disorders [12,13]. For instance, up to one-third of post-TBI patients have been diagnosed with convulsive disorders at a 15 year follow-up interval [14,15]. This high prevalence suggests that, in addition to axonal injury, damage to the cortical regions may lead to pathological changes at the neuronal synaptic level, which are critical for regulating communication across neural networks and may contribute to seizure development.

Synaptic dysfunction and loss in traumatic brain injury may be incurred either acutely or chronically. In an acute trauma, the shearing impact of mechanical injury can lead to disintegration of both presynaptic and postsynaptic neuronal terminals. Animal studies have reported evidence of dendritic spine fragmentation in the hippocampal region following the application of a controlled cortical force [16]. Synaptic dysfunction has also been observed even after the application of mild trauma to the cortical tissues in mice, despite minimal cell death [17]. Beyond hippocampus, early dendritic spine degeneration is also evident in the prefrontal cortex and thalamus, where damage to these neuroanatomical structures is linked to executive dysfunction and sensory deficit [18]. In addition to dendritic degeneration and loss of synaptic densities, diffuse axonal injury (DAI) can also result in varicosities that peak around 24 h post-injury [19,20]. Moreover, studies have indicated that when an axon is transected, the presynaptic region is disintegrated almost immediately, while the postsynaptic spine later undergoes retraction [21]. The structural synaptopathy of TBI significantly evolves over time. Indeed, weeks or months following an acute injury, additional synaptic degeneration may occur via activation of proinflammatory pathways and microglial pruning—changes that may underlie ongoing memory deficits post-TBI [22,23].

Besides structural synaptopathy, TBI has also been associated with functional alteration of neuronal synapses. Key post-TBI events include disruption of the excitatory/inhibitory (E/I) equilibrium, metabolic disruption, dysregulation of network connectivity and impaired synaptic plasticity [24,25,26]. To further elaborate on the E/I imbalance, the post-TBI phase involves a significant surge in glutamate secretion, coupled with impaired glutamate clearance. This causes excessive activation of NMDA (N-methyl D-aspartate) and AMPA (α-amino-3-hydroxy-5-methyl-4-isoxazolepropionic acid) receptors on postsynaptic membranes, leading to calcium ion (Ca^2+^) influx and subsequent neuronal damage [27]. Moreover, impaired gamma-aminobutyric acid (GABA) signaling in TBI reduces inhibitory synaptic currents and delays the maturation of inhibitory pathways [28]. The net result is an E/I imbalance favoring excitation in the acute phase, which has been corroborated in a study where whole-cell recordings in mouse piriform cortex revealed a robust increase in excitatory synaptic activity 1 h after the primary brain insult [21]. Impaired synaptic plasticity is another functional hallmark of synaptopathy in TBI. This often manifests as an aberrant long-term potentiation (LTP), where the latter is a phenomenon characterized by an increase in synaptic efficacy. This correlates with a decline in hippocampal memory formation [29].

From the practical perspective, understanding the neuropathogenesis of TBI-induced synaptic excitotoxicity is essential for developing effective treatment protocols. Synaptic dysfunction is correlated with the spectrum of clinical symptoms found in TBI survivors. Disruption of E/I balance and impaired synaptic plasticity contribute to deficits in long-term potentiation and memory encoding in the hippocampus [25]. Comparably, glutamatergic toxicity can also provoke network hyperexcitability, predisposing individuals to post-traumatic epilepsy [30,31]. Furthermore, diminished GABAergic tone and delayed maturation of inhibitory circuits may also underlie emotional dysregulation and sleep disorders [32]. Most notably, synaptic loss in prefrontal and limbic circuits has been associated with executive dysfunction and depression [33]. Keeping this in mind, interventions aimed at synapse protection and regeneration offer promising potential treatment. In line with this, Sloley S.S. et al., 2021 [34] showed that pre-treatment of mice with memantine (an NMDAr antagonist) prevented the maladaptive synaptic and transcriptional changes caused by repetitive brain injury, mediated through glutamatergic excitotoxicity. Conversely, post-injury administration of memantine has also been reported to reduce tau phosphorylation and partially preserve hippocampal LTP in experimental TBI models [35]; Table 1 provides an overview of outcomes reported across studies on TBI-related interventions. In the diagnostic realm of TBI, synaptic alterations also search for biomarkers and neuroimaging tools [36,37]. Advanced imaging methods are being developed to measure synaptic density while synaptic proteins are being evaluated as fluid biomarkers in TBI [19].

The goal of this review is to synthesize current evidence on how TBI alters the synaptic structure and function across the various post-trauma phases. By integrating findings from preclinical models and human studies, we elaborate how synaptic pathology evolves over time and how various synaptic elements can be manipulated for improving post-TBI outcomes [49].

## 2. Post-TBI Synaptic Pathology—Mechanistic Insights into Chronology of Post-TBI Synaptic Changes

The post-TBI phase involves a cascade of time-specific synaptic events encompassing the acute, subacute and chronic periods. The earliest phases are characterized by glutamatergic excitotoxic injury, with oxidative stress and neuroinflammation being the hallmarks of the subacute phase [50,51]. Lastly, the chronic phase features neuronal reorganization and tau pathology [52].

### 2.1. Acute Phase: Excitotoxicity

In the immediate aftermath of an impact to the brain, mechanical stretch and cell membrane disruption precipitate massive ionic flux. Voltage-gated Ca^2+^ channels (VGCCs) become dysregulated and can even be physically damaged, thereby allowing the presynaptic terminals to spontaneously secrete glutamate into the synaptic cleft [53,54]. Supporting this, administration of the VGCC blocker omega-conotoxin has been effective in downregulating Ca^2+^ levels in an in vitro model of Diffuse Axonal Injury (DAI) [54]. Patients with severe TBI also demonstrate surges in cerebrospinal fluid (CSF) glutamatergic levels where high glutamate has been associated with poor clinical outcomes [55,56]. At the structural level, acute excitotoxic loss is most prominent in hippocampal CA1 pyramidal neurons and cortical layers II/III, regions that sustain early synaptic disintegration [57].

An important neurophysiological mechanism underlying acute excitotoxicity in TBI is cortical spreading depolarization (CSD). CSD primarily involves a rapidly propagating wave of depolarization which spreads across the neuronal and glial cells and involves an aberrant movement of ions across the neuronal membranes. This is characterized by high extracellular concentrations of K^+^ ions and glutamate [58,59]. Glutamate binds to postsynaptic NMDA and AMPA receptors, inducing a rapid Ca^2+^ influx which triggers downstream activation of kinases, proteases and nitric oxide synthase. This subsequently results in cleavage of cytoskeletal and synaptic proteins, mitochondrial disruption and free-radical production, which contribute to the loss of dendritic integrity (Figure 1). Critically, the acute secondary injury phase is characterized by profound metabolic imbalances that compromise cellular energy metabolism [11,60]. Mitochondrial dysfunction, triggered by excessive Ca^2+^ influx and oxidative stress, impairs oxidative phosphorylation and disrupts adenosine triphosphate (ATP) production within hours of the initial insult [61,62]. This state of "metabolic crisis" can occur even in the absence of ischemia, indicating primary mitochondrial failure rather than substrate limitation [61]. These acute metabolic perturbations exacerbate the cascade of neuronal damage and represent critical therapeutic targets for early intervention strategies [63]

As a biomarker of excitotoxicity-induced cytoskeletal damage, immunoreactivity results have elevated CSF levels of SBDP145 (αII-spectrin breakdown products) where the latter is characterized as a major breakdown product derived through cleavage of the cytoskeletal protein α-II-spectrin [64]. Rapid modulation of AMPAr is also a notable event during the acute phase. The postsynaptic AMPAr comprises four subunits where GluR2 is responsible for regulating transmembrane Ca^2+^ movement. Following the neuronal insult, Ca^2+^-mediated activation of protein kinase C (PKCα) boosts postsynaptic membrane insertion of AMPAr devoid of GluR2 [65]. This acts as a “feed-forward” loop to boost Ca^2+^ ion entry into the postsynaptic neuron, thereby further accentuating neurotoxicity [66]. Calcium secretion is also paired with increased activity of the synaptosome-associated protein (SNAP-25) and syntaxin, major protein components of the SNARE complex, which mediate the presynaptic release of Ca^2+^ ions [50], as indicated in Figure 1.

**Figure 1 ijms-26-09604-f001:**
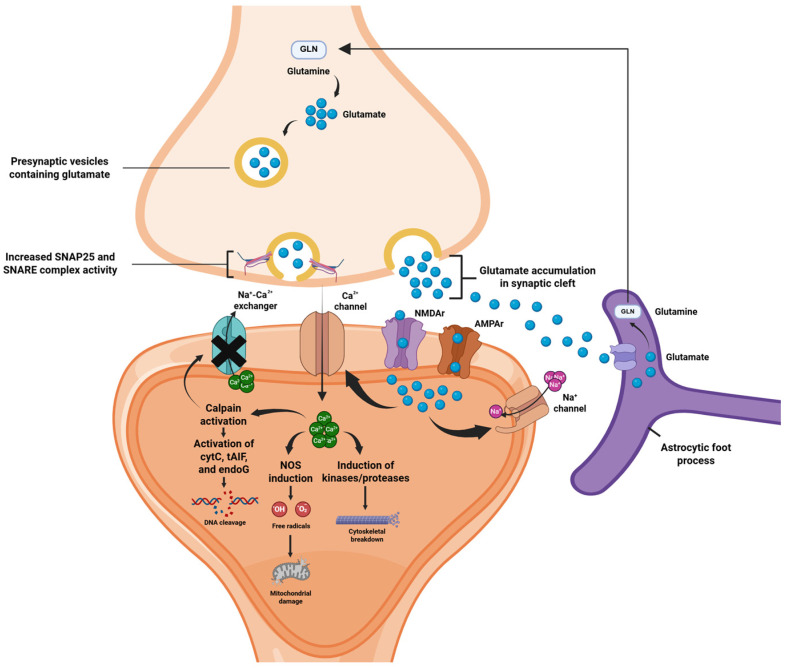
Acute TBI and Excitotoxicity. Main excitotoxic events in acute TBI include increased activity of the SNARE proteins supporting increased glutamatergic exocytosis and binding at the postsynaptic NMDAr and AMPAr, along with increases in the postsynaptic Ca^2+^ and Na^+^ ion influx. Excess glutamate is recycled by neighboring astrocytes. Downstream activation of Ca^2+^-mediated enzyme pathways leads to cytoskeletal breakdown and further boosts the production of reactive oxygen species, thereby potentiating mitochondrial damage. Moreover, Ca^2+^-mediated calpain activation leads to the activation of cytochrome c (cytC), truncated apoptosis-inducing factor (tAIF) and endonuclease G (endoG), which are implicated in DNA lysis. Calpain also blocks the Na^+^-Ca^2+^ exchanger, thereby limiting outflow of Ca^2+^ ions and further potentiating intracellular calcium toxicity [67].

As early as 24 h following a controlled cortical injury (CCI) in a preclinical TBI model, pyramidal neurons have been shown to incur a loss of up to 20–32% of dendritic spines within both the ipsilateral and contralateral cerebral hemispheres [68]. Moreover, synaptophysin immunostaining in hippocampus shows the loss of approximately one-third of presynaptic puncta within the first week of brain injury [17]. Together, these findings highlight the rapid and widespread synaptic degeneration that occurs in the acute phase following TBI, likely contributing to the cognitive and behavioral impairments observed after injury.

### 2.2. Subacute Phase: Neuroinflammation

Over days to weeks of synaptic dysfunction following TBI, neuroinflammatory cascades begin to dominate the pathological process. This is characterized by the activation of microglia, as well as dendritic cells and astrocytes (Figure 2). Microglia have been selectively shown to engulf presynaptic terminals of excitatory neurons for weeks post-TBI [22]. Concomitantly, elevated synaptic glutamatergic activity can linger for one week [19] and during this interval, both NMDA and AMPA receptors continue to mediate a pathological influx of Ca^2+^ ions. Prolonged depolarization also promotes the mitochondrial generation of reactive oxygen species (ROS) and lipid peroxidation, thereby promoting an “oxidative stress” environment. As inflammation and signaling cascades progress further, residual spines undergo thinning and incur loss of their mature mushroom morphology [17]. Ongoing microglial phagocytosis, coupled with a lack of new spine formation, suggests there is no net recovery of lost synapses during the subacute timepoints. In line with this, synaptophysin staining in hippocampal dentate gyrus reveals only a minimal terminal regrowth during the early stages of synaptic injury [69,70]. Astrocytes can also proliferate (astrogliosis) and form a glial scar around injury cores, further isolating the damaged synapses [71,72]. Further, elevated CSF levels of IL-6 (a marker of inflammation) are often correlated with poor global outcomes in the post-TBI period [73] (Figure 2).

The neuroinflammatory phase is also marked by downregulation of inhibitory GABA-A receptors. In line with this, the δ subunit of GABA-A receptors has been shown to be reduced in the dentate granule cells by 44% during week 1, with the effect persisting longer than 15 weeks. Moreover, GABA-B receptor subunits and tonic currents were also markedly reduced [76]. At the excitatory synapses, NMDA receptor subunit composition may undergo dynamic changes. Concomitantly, animal data show increases in NMDA receptor subunit 2B (NR2B) up to months after injury, which also serves to prolong Ca^2+^ ion influx. Loss of inhibitory interneurons marks another major sequela in terms of boosting excitotoxic injury, with evidence suggesting that up to 70% of parvalbumin (PV)-expressing fast-spiking interneurons may become damaged in the peri-lesional cortex. Neuroinflammatory pruning is particularly pronounced within the dentate gyrus and peri-lesional cortex [77]. Collectively, these alterations reflect a profound and long-lasting imbalance between excitatory and inhibitory signaling in the post-injury brain, creating a neurochemical environment that promotes network hyperexcitability and increases vulnerability to further damage.

### 2.3. Chronic Phase: Network Remodeling

The chronic phase following TBI is characterized by persistent synaptic remodeling and long-lasting circuit-level dysfunction [78]. While synaptogenesis may occur post-TBI [79], many studies have reported sustained reductions in synaptic densities. In rats, hippocampal spine density remains severely low for one week after severe TBI but starts rising until the numbers plateau approximately up to four weeks after the initial insult [19]. The chronic post-TBI phase is marked by persistent E/I imbalance, characterized by deficits in GABAergic neurons and a chronic increase in NMDAr activity, which maintains the substantially elevated Ca^2+^ ion influx [27]. Combined with decreased inhibitory drive, this creates a pro-epileptogenic milieu which is consistent with post-traumatic epilepsy. Additionally, hippocampal LTP is also impaired where moderate-severe TBI patients report LTP deficits at 2–30 days post-injury [24]. These deficits often persist in the chronic phase. Postsynaptic density protein PSD-95 is markedly reduced in the hippocampus after TBI, and synaptic vesicle proteins (e.g., SV2A) may remain depleted in hippocampus and cortex weeks after injury [24].

Another important hallmark of chronic TBI features neuropathological accumulation of amyloid precursor protein (APP) within the axonal spheroids, peaking at 24 h following TBI. Neuronal accumulation of APP can underlie chronic axonal transport deficits [80,81]. Over longer periods, pathological tau spreads through affected networks, contributing to the pathogenesis of chronic traumatic encephalopathy (CTE) pathology [82]. The synaptic connectivity is permanently altered, thereby causing many excitatory synapses to become miswired. These changes underlie persistent functional deficits, e.g., memory impairment [83], mood disorders [84] and seizures [85], often observed in TBI survivors.

## 3. Synaptic Impairment and Psychiatric Comorbidities

In addition to neurological sequelae, psychiatric and neurobehavioral symptoms are common in the aftermath of TBI. Brain trauma, whether mild, moderate or severe, has been recognized to be associated with mood instability, precipitating apathy, agitative behaviors and suicidal ideation [86,87]. Such psychiatric sequelae, when combined with cognitive impairment, can potentiate a significantly poor functional outcome among TBI survivors [88]. Additionally, rodent TBI models often elicit post-injury anxiety and depression-like behaviors, particularly when combined with psychosocial stressors [89]. Furthermore, the efficacy of selective serotonin reuptake inhibitors (SSRIs) and serotonin–norepinephrine reuptake inhibitors (SNRIs) in alleviating post-TBI depressive symptoms suggests that these psychiatric effects may share underlying neurochemical mechanisms with primary mood disorders [90,91].

Another common link between major depressive disorder (MDD) and post-TBI depression is the disruption of glutamatergic signaling. An altered activity of glutamate transporters has been established as a potential mechanism underlying the pathogenesis of MDD [92]. In line, ref. [93] have also hinted at a decreased glutamatergic uptake at the synapses mediated by downregulation of glutamate transporter proteins (GLT-1 and GLAST). This results in an increased extracellular glutamate activity which is synergistic with post-TBI excitotoxicity and depressive behavior. In addition, injury-induced release of cytokines, e.g., tumor necrosis factor (TNF-α) and interleukin-1β (IL-1β), remains chronically elevated for weeks to months post-TBI. These proinflammatory molecules can further alter glutamate/GABA balance, receptor trafficking and neuronal excitability [94].

The association between synaptic injury and post-traumatic stress disorder (PTSD) is also well-documented. Chronic stress animal models indicate a reduction in the overall expression of brain-derived neurotrophic factor (BDNF) and postsynaptic density protein 95, and downregulation of AMPA GluR1 subunit, which can trigger synaptic dysregulation [95]. Furthermore, dendritic atrophy and loss of glutamatergic synapses in the hippocampus and prefrontal cortex (PFC) are also noted in animal models of PTSD. Such synaptic pruning undermines the function of the limbic network, thereby impairing cortical functions of emotional and behavioral regulation [96]. Widespread lower synaptic density has been demonstrated among mice models of repetitive mild TBI and chronic stress, which is also marked by an increase in regional synchrony to compensate for synaptic damage [97]. Interestingly, hippocampal neuroinflammation is found to be the most severe for PTSD animal models who undergo a subsequent head injury. This interconnection holds significance for the patients with pre-existing PTSD, since they might be more prone to the neuropsychiatric sequelae of TBI [98]. Using positron emission tomography (PET), and the radiotracer [^11^C]UCB-J, which binds to the synaptic density marker synaptic vesicle protein (SV2A), lower synaptic density in mood-related circuitry was associated with higher depression severity across patients with MDD and/or PTSD [99]. Taken together, these preclinical findings highlight a shared pattern of synaptic vulnerability across chronic stress, PTSD and TBI models—particularly within mood and emotion-regulating circuits. Translating these insights into human studies through neuroimaging tools such as PET, especially with SV2A ligands like [^11^C]UCB-J, offers a promising approach to quantifying synaptic loss and understanding its role in neuropsychiatric outcomes following trauma.

## 4. Synaptic Pathology in TBI—Therapeutic Implications

Understanding the therapeutic implications of synaptic dysfunction in TBI is crucial for developing effective interventions.

An important mechanism underlying post-TBI excitotoxicity is the unchecked glutamatergic signaling. This renders the modulation of glutamatergic pathway a major target in terms of improving TBI-related outcomes. Accordingly, the NMDAr antagonist memantine has been tested in animal TBI models. In a mouse model of repetitive mild TBI, ref. [35] treated animals with memantine (10 mg/kg) and found molecular benefits, but nil behavioral gain was reported. Memantine-treated mice showed reduced tau phosphorylation and lower astrogliosis during the acute stages of synaptic pathology. Furthermore, deficits in hippocampal LTP were also improved up to one month post-injury. Another recent preclinical trial administered 10 mg/kg memantine daily, following repetitive closed-head injury in rats. Memantine strongly inhibited pathologic propagation of CSD, whereas memantine-treated rats also scored higher in terms of neurobehavioral testing [38].

Another glutamate-related intervention is the use of β-lactam antibiotic ceftriaxone, which upregulates the GLT1 glutamate transporter. The latter is physiologically implicated in the synaptic uptake of glutamate, while TBI significantly leads to the downregulation of GLT1. In their study, [39] administered ceftriaxone (200 mg/kg/day for 7 days) to rats after fluid-percussion TBI. Ceftriaxone restored cortical GLT1 levels and prevented the progressive loss of GABAergic inhibition. It also preserved parvalbumin-expressing interneuron markers, further reflecting reduced glutamate excitotoxicity. Although this study did not report cognitive scores, the molecular outcomes imply that boosting the glutamate uptake has a definitive neuroprotective role. Kv7 (M-type) K^+^ channels also modulate the excitability downstream of glutamate signaling. Ref. [40] tested an M-channel opener retigabine in a mouse model of blast injury. Acute retigabine dosing after each blast shock sharply reduced post-traumatic seizures. In retigabine-treated mice, the duration of acute seizures was significantly shorter, and chronic spontaneous seizures were nearly abolished. These results suggest that enhancing hyperpolarization and thus decreasing the glutamate-driven neuronal firing can mitigate synaptic hyperexcitability after TBI.

Comparably, synaptic modulation of GABAergic neurons can also play a potential role in modulating TBI outcomes. In their study, ref. [41] used a novel drug (522-054) that allosterically enhances α7-nicotinic acetylcholine receptors (α7-nAChr) while inhibiting α5-containing GABA-A receptors. In a TBI mouse model, they employed systemic treatment with 522-054 rescued hippocampal synaptic function. Hippocampal slices obtained from treated rats showed an improvement in CA1-LTP, as compared to deficits in untreated TBI rats. Treated mice also performed better during cognitive tasks (improved fear memory and Morris water maze performance) in the chronic phase, despite no changes in gross cortical tissue atrophy [41]. This study demonstrates that modulating GABAergic tone and cholinergic drive up to 12 weeks post-injury can restore synaptic plasticity and memory. Similar outcomes were obtained by [42], who utilized AVL-3288, another positive modulator of α7-nAChr. A consistently positive trend was noted in terms of cue and fear memory. Moreover, hippocampal atrophy was also found to be reduced, with increments noted in the LTP within the CA1 region.

Another promising approach is cell therapy. Ref. [43] transplanted embryonic medial ganglionic eminence (MGE) GABAergic progenitor cells into the adult mice brain one week after a controlled cortical impact (CCI). The transplanted interneurons migrated into the hippocampus and differentiated into mature GABAergic neurons, and increased inhibitory synaptic currents in host circuits. From a behavioral perspective, TBI mice with transplanted GABAergic interneurons showed dramatic improvements in spatial memory when compared to controls. Furthermore, MGE-treated mice also had fewer spontaneous seizures during the period of chronic recovery [43]. This aspect also renders them as potentially effective in epileptiform disorders [100]. Clinically, enhancing GABAergic function has been tested in TBI patients. In one study, TBI patients were randomized to receive either a GABA analog, gabapentin (300 mg twice daily), or a placebo for two weeks. The gabapentin group showed significantly greater improvement in Glasgow outcome scores, along with fewer episodes of paroxysmal sympathetic hyperactivity. A significantly lower mortality was also reported in the gabapentin group [44].

Ketamine has also gained interest since it can blunt the glutamate-driven excitotoxicity while also promoting synaptic plasticity. Importantly, clinical and preclinical data have indicated that ketamine is safe to administer in head-injury settings and can potentiate rapid anti-depressant effects in TBI [99]. In rodent TBI models, subanesthetic dosing of ketamine has been shown to reduce neuroinflammation without worsening behavioral recovery [101]. Functionally, ketamine has been shown to improve post-injury synaptic function and memory. In line, intravenous ketamine infusion (10–20 mg/kg IV) after closed-head injury increased synaptic puncta density in the medial prefrontal cortex (mPFC) at four days post-TBI [102]. In mice given a cortical impact injury, one week of systemic ketamine boosted microglial cell proliferation and reduced numbers of astrocytes. Moreover, an improved performance in spatial learning tasks was also demonstrated, despite a reduction in excessive hippocampal proliferation [103]. These cognitive gains imply enhanced circuit function, even where neurogenesis was suppressed. In clinical trials, a therapeutic effect of ketamine has also been elicited through its ability to boost axonal densities amongst patients with lower baseline SV2A activity which, in turn, is also associated with clinical improvement in depression [104].

Beyond glutamate and GABA, other synaptic modulators may also promote post-TBI recovery. Serotonergic agonists (5-HT1A and 5-HT2A) have been shown to be effective in terms of boosting the neuronal activity of BDNF in response to acute stress [105]. In line with this, ref. [45] tested the 5-HT_6_R agonist, classified as WAY-181187, which is known to upregulate BDNF [106]. In a rat CCI model, five daily doses of WAY-181187 (3 mg/kg), starting acutely post-injury, significantly improved spatial learning and memory. Concomitantly, treated TBI rats showed higher Morris water maze scores at one- and four-week intervals. At the molecular level, WAY-181187 increased BDNF expression within the hippocampus and prefrontal cortex while also boosting the mushroom-shaped dendritic spine densities within the hippocampal region. This study indicates that facilitating neurotrophic support can potentially heal damaged neuronal circuitry after TBI [45]. In addition, anti-inflammatory chemicals have also been utilized in preclinical trials. Co-administration of N-acetylcysteine (NAC) and minocycline has been shown to preserve synaptic function and offer neuroprotective effects for mice models with a closed head injury while improving mice performance in spatial memory-related tasks [46]. Anti-oxidative potential of erythropoietin (EPO) has also been investigated due to its combined anti-inflammatory and anti-glutamatergic potential. In line with this, ref. [47] found that exogenous EPO administration to post-TBI mice helped reduce neurodegenerative changes and boost synaptic density within the hippocampal and limbic nuclei at six month intervals. The effect of systemic steroids has also been explored, where dexamethasone has been noted to have sex-specific effects in TBI animal models. Following an acute CCI, a dexamethasone injection can potentially allow male mice to experience a greater reduction in injury volume, as well as a lower extent of astrogliosis, cytokine-mediated neuroinflammation and microglia activation [48].

## 5. Conclusions and Future Directions

Growing evidence converges on a model in which synaptic dysfunction and loss in trauma are driven by overlapping mechanical and biochemical cascades. The initial insult and ensuing excitotoxic cascade increase the extracellular glutamate levels while downregulating glutamate reuptake. This is coupled with a disproportionate injury encompassing GABAergic interneurons, which tips the cortex towards hyperexcitability. While acute and subacute phases in TBI exhibit mismatched E/I balance, the chronic phase of neuroinflammation sustains lasting synaptic deficits. This synaptopathy, characterized by dendritic spine loss, neurotransmitter imbalance and impaired synaptic plasticity, underlies the neurocognitive, behavioral and epileptiform sequelae of TBI.

Looking forward, emerging biomarkers could play a pivotal role in further strengthening the correlation between synaptic densities and TBI. Both fluid and imaging markers of synaptic integrity could potentially enable early synaptic pathology detection and patient stratification. In addition, as we further explore emerging synaptic therapies in both acute and chronic TBI, redefining the therapeutic window remains essential where clinically relevant regimens should ideally retain efficacy with 12+ hour delays for severe TBI and even prolonged delays in mild cases of neuron injury [107,108]. To effectively translate these mechanistic insights from animal models to human patients, advanced neuroimaging tools are critical. PET imaging using synaptic density markers such as [^11^C]UCB-J, which binds to synaptic vesicle glycoprotein 2A (SV2A), enables in vivo quantification of synaptic loss across brain regions implicated in mood, cognition and sensory-motor control. In parallel, functional MRI (fMRI) and network-level analyses of functional connectivity offer valuable insights into the large-scale reorganization of brain networks following TBI. Together, these approaches can help characterize how synaptic integrity and circuit function evolve over time in the human brain. Importantly, a phase-specific understanding of acute, subacute and chronic changes in synaptic function is essential to identify therapeutic windows and guide interventions aimed at restoring brain function. Such efforts could ultimately inform treatments, and perhaps even preventative strategies, for the constellation of persistent post-concussive symptoms that affect many individuals after a head injury, even in cases of mild TBI.

Furthermore, the development of effective interventions will likely require a combination of targeted strategies that modulate glutamatergic, GABAergic and neurotrophic signaling pathways in parallel. Cell-based therapies may offer additional synergistic benefits by promoting synaptic repair and circuit regeneration. As our ability to map synaptic changes in the human brain continues to advance through tools like SV2A PET and MRI measures of structural and functional connectivity, there is growing potential to tailor multi-modal therapeutic approaches to individual patients and injury profiles. Ultimately, integrating mechanistic insights with translational biomarkers and phase-specific interventions may pave the way for restoring synaptic function and preventing long-term neuropsychiatric and cognitive sequelae following TBI.

## Figures and Tables

**Figure 2 ijms-26-09604-f002:**
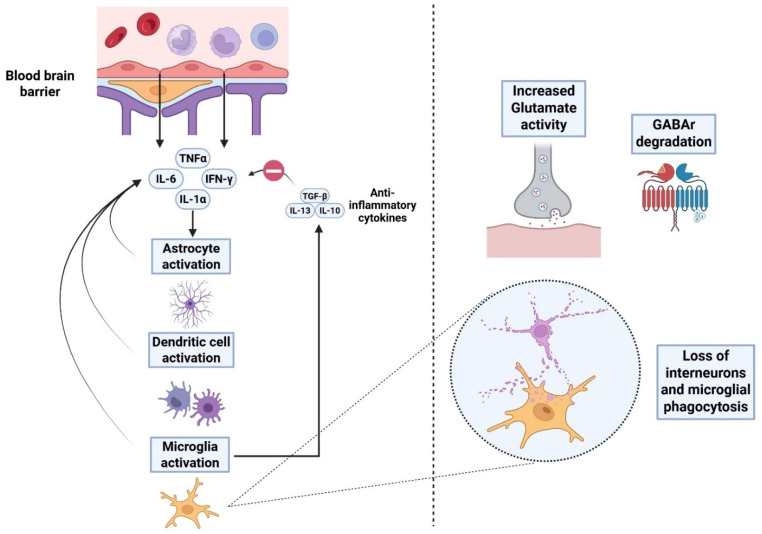
Neuroinflammation as a result of TBI. Proinflammatory cytokines leak through the blood–brain barrier and initiate activation of microglia and dendritic cells, thereby potentiating astrocytic proliferation and astrogliosis. This further boosts the secretion of proinflammatory cytokines (tumor necrosis factor: TNF-α, interleukins-1α, interleukin-6 and gamma-interferon) (IFN-γ). Further highlights of the subacute phase in TBI include increased glutamatergic activity, loss of inhibitory interneurons and breakdown of GABA signaling. Moreover, production of anti-inflammatory cytokines also occurs to resolve neuroinflammation [74,75].

**Table 1 ijms-26-09604-t001:** Review of outcomes from various studies focused on TBI-related interventions.

Study Authors	Participants	Experimental Group	Control Group	Main Intervention	Study Outcomes
[35]	Male rats, mean age of 8 weeks (n = 95)	rm-TBI (4 injuries in 4 days) treated with memantine, n = 29	rm-TBI (4 injuries in 4 days and placebo treatment), n = 37 Sham injuries in 4 days, n = 30	Memantine (NMDA antagonist) given within 1 h after the last mTBI (10 mg/kg)	Reduced tau phosphorylation and less glial cell activation; restored hippocampal LTP at 1 month; no major improvement noted in behavioral outcomes.
[38]	Male rats, mean age of 9 weeks (n = 31)	rm-TBI (4 injuries in 4 days) treated with memantine, n = 15	rm-TBI (4 injuries in 4 days and placebo treatment), n = 16	Memantine (10 mg/kg) after the first TBI and prior to the second, third and fourth impacts	Inhibited cortical spreading depolarizations and reduced post-depolarization oligemia; improved neurobehavioral scores.
[39]	Male rats, mean age of 12 weeks (n = 21)	Fluid percussion injury treated with ceftriaxone, n = 7	Saline-sham, n = 7 Saline-TBI, n = 7	Ceftriaxone (β-lactam antibiotic, 250 mg/kg), once daily for 7 days	Reduced post-TBI GLT-1 downregulation; preserved intracortical inhibition by electrophysiology and parvalbumin interneuron markers.
[40]	Male rats, mean age of 12 weeks (n = 23)	Repetitive blast injury with retigabine treatment, n = 12	Blast group, n = 6 Sham group, n = 5	Retigabine (M-channel opener), 1.2 mg/kg dose, administered 30 min after each blast exposure	Reduced duration of acute post-traumatic seizures; markedly decreased development of chronic epilepsy post-injury.
[41]	Adult male rats (n = 45)	Fluid percussion injury with 522-054 treatment	Sham-vehicle Sham-522-054 TBI-vehicle	522-054 (α7-nAChr agonist + α5-GABA-A inhibitor)	Restored hippocampal LTP; improved fear memory and water-maze performance; no changes in hippocampal or cortical atrophy.
[42]	Male rats, mean age of 2–3 months (n = 94)	Fluid percussion injury with AVL-3288 treatment, n = 11	Sham-vehicle, n =10 Sham-AVL-3288, n = 8 TBI-vehicle, n = 11	A total of 8 treatments completed for AVL-3288 (0.3 mg/kg), a α7-nAChr agonist	Improved performance in water maze retention and working memory, better cue and contextual fear memory, reduction in hippocampal atrophy.
[43]	Adult male mice	Controlled cortical impact (CCI) with embryonic MGE transplant, n = 11	Sham mice, n = 16 TBI only mice, n = 19	Transplanted embryonic MGE-derived GABAergic interneurons	Grafted interneurons integrated and increased synaptic inhibition; improved spatial memory precision; reduced seizures.
[44]	RCT with TBI patients in ITU (n = 65), moderately decreased GCS (8–13) and severely decreased GCS (<8)	TBI patients receiving gabapentin, n = 30	TBI patients receiving placebo, n = 30	Administration of gabapentin, a GABA analog, 300 mg twice daily for 2 weeks	Significant improvements noted in Glasgow outcome scale and GCS measures up to 90 days after trauma.
[45]	Male rats, mean age of 6–7 weeks	Moderate CCI with administration of WAY-181187	N/A	WAY-181187 (5-HT6R agonist), 3 mg/kg for 5 days	Improved Morris water maze performance at 1 and 4 weeks; increased hippocampal BDNF and improved dendritic spine density.
[46]	Male rats, mean age of 16–18 weeks	Closed-head injury with administration of minocycline/NAC, n = 5	Sham injury-saline, n = 5	Minocycline (22.5 mg/kg) plus NAC (75 mg/kg) 3–5 days after injury	Improved performance in Barnes maze and hippocampal neuroprotection, and decreased synaptic loss in TBI.
[47]	Male rats, mean age of 6–8 weeks	CCI with EPO administration, n = 15	Sham-EPO, n = 15 CCI-vehicle, n = 15	Up to 5000 U/kg EPO, total of 6–9 doses	Improved cued-fear memory response along with increased synaptic density in the hippocampus and amygdala.
[48]	Male and female rats, mean age of 12 weeks	CCI with left-sided moderate TBI, administered Lipo-Dex, n = 5	Lipo alone, n = 5	Up to 3 mg/kg of dexamethasone given 1 h post-CCI	Male mice experienced significantly greater reduction in lesion volume, astrogliosis, cytokine-mediated inflammation and microglia activation when compared to female mice.

rm-TBI: Repetitive mild traumatic brain injury; NMDA: N-methyl D-aspartate; LTP: Long-term potentiation; GLT: Glutamate transporter; nAChr: N-acetylcholine receptor; GABA: Gamma-amino butyric acid; MGE: Medial ganglionic eminence; GCS: Glasgow coma scale; BDNF: Brain-derived neurotrophic factor; NAC: N-Acetylcysteine; EPO: Erythropoietin; Lipo-Dex: Liposomal carrier-dexamethasone; RCT: randomized controlled trial; CCI: controlled cortical impact; N/A: not applicable.

## Data Availability

No new data were created or analyzed in this study. Data sharing is not applicable to this article.

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
