# Peer review of "Synaptic Pathology in Traumatic Brain Injury and Therapeutic Insights"

_ijms, 2025, doi:10.3390/ijms26199604_

Round 1

Reviewer 1 Report

Comments and Suggestions for Authors

The review examines the acute, subacute and chronic phases of synaptic structure and function disorders that occur after traumatic brain injury. The loss of dendritic spines and disruption of synaptic plasticity are undoubtedly included in the main causes of pathological changes that follow such effects - therefore, the review deserves attention.

The authors propose to consider the analysis of disorders occurring after injury in three periods - during the acute phase, subacute and chronic course of processes, which is an advantage of the proposed review. I agree with the authors that during the acute phase, the main process is excitotoxicity, but it would be good to emphasize that this does not exhaust the damage to the nervous tissue, the authors in the text indicate such processes, for example, edema and mitochondrial dysfunction, this should be indicated in the abstract. The same – it is necessary to emphasize that in the subacute phase, in addition to ongoing glutamate excitotoxicity and GABAergic disorders, along with neuroinflammatory pathology, other processes occur, such as autophagy. In the chronic phase, long-term structural remodeling of neuronal networks is observed. These chronic changes underlie persistent cognitive impairment and memory deficit, mood disorders and the development of post-traumatic epilepsy.

GENERAL COMMENTS:

The scientific significance of the review would be significantly higher if the authors analyzed in which brain structures the changes under consideration were observed.

It is necessary to indicate which brain structures suffer first after the injury, since when citing articles, in most cases, the authors do not indicate where the disorders occur (the hippocampus is often indicated, but it should be understood that the hippocampal neurons are the most vulnerable to any damaging effects).

The chronic phase is characterized too briefly. It is necessary to supplement it with data on molecular markers of the state of synapses (synaptic vesicles, post-synaptic densities). In addition, when the authors indicate the neurological consequences of the injury, indicating the brain structures associated with individual symptoms, neurotransmitter specificity and the effectiveness of pharmacological agents will indicate more specific reorganizations of neural networks. The same applies to cognitive functions - there are good models of neural circuits that provide such functions.

When describing the pharmacological correction of disorders, the authors do not indicate in what period drugs were used (before or after the injury).

MINOR COMMENTS:

Line 64:

In an acute trauma, the shearing impact of mechanical injury can lead to disintegration of both presynaptic and postsynaptic neuronal axons. (Synapses are not only axo-axonal).

Line 94

Understanding the neuropathogenesis of TBI-induced synaptic toxicity is essential for developing effective treatment protocols. (Can a synapse be toxic?)

Line 142

This subsequently results in cleavage of cytoskeletal and synaptic proteins, mitochondrial disruption and free radical production which contribute to loss of dendritic integrity (Figure 1).

This is not shown in the figure, but I would like it to be so.

Notes on the figures.

Fig. 1. Although the figure is intended to explain excitotoxicity, it does not do so. The understandable desire to point to SNAP25 as the main mediator between traumatic impact and excitotoxicity has oversimplified the overall picture. Excitotoxicity involves many more participants and components, for example, extrasynaptic ionotropic and metabotropic receptors. At a minimum, astrocytic processes should be shown to remove excess glutamate. In order not to overload the text, relevant references should be provided.

In addition, the figure may lead one to think that apoptosis and cell swelling occur in the postsynaptic compartment.

Erroneous labeling of glutamatergic vesicles (should be glutamate-containing).

Fig. 2. Proinflammatory cytokines penetrate the damaged BBB, but are subsequently synthesized by many brain cells, and in response, the synthesis of anti-inflammatory cytokines is activated to resolve neuroinflammation. This important component should be mentioned, along with mitophagy and autophagy, as they serve as targets for therapy.

In addition, given that the authors' main focus in the review is on synaptic changes, it is necessary to point out the participation of microglia in the removal of damaged dendritic spines and synaptic components.

Reviewer 2 Report

Comments and Suggestions for Authors

Review of the manuscript entitled: “Synaptic pathology in Traumatic Brain Injury and Therapeutic Insights” authored by Poojith Nuthalpati, Sophie E Holmes, Hamada H. Altalib and Arman Fesharaki-Zadeh

Thank you for possibility to review this interesting manuscript.

Presented manuscript about synaptic pathologies occurring after traumatic brain injury (TBI) undertakes important scientific and clinical problem of modern world.  Manuscript is enhanced by two very clear figures and one table. Information presented in the review are logically organized and are supported by recent literature (81 papers are from the last 10 years). Overall, authors’ work has some flaws that should be corrected before manuscript is published.

Major concerns:

  • Figure 1 & text – authors consider only apoptosis as the main pathway of cell-death after TBI which of course is true. However, necrosis pathway is also very important in terms of TBI-induced excitotoxicity (for example doi:10.1016/j.csbj.2015.03.004)
  • Figure 2 & text – authors mention about leakage of cytokines from bloodstream to brain through unsealed blood-brain barrier. However, after TBI astrocytes or microglia are able to secrete pro-inflammatory cytokines and neurotoxic molecules by themselves. This issue is not clearly presented in figure, neither in text (for example doi: 10.1016/j.ibneur.2025.07.009).

Minor concerns:

  • Some typos and punctuation errors
  • Inconsistency with abbreviations (for example LTP is not introduced)
  • Figure 2 is a little bit blurry in my version of the manuscript

Round 2

Reviewer 1 Report

Comments and Suggestions for Authors

The authors have made corrections, and the figures are now more informative. I've noted a minor error: The first column of the table lists references, but states the authors of the works. I believe the manuscript can be published.